# PRG3 and PRG5 C-Termini: Important Players in Early Neuronal Differentiation

**DOI:** 10.3390/ijms232113007

**Published:** 2022-10-27

**Authors:** Nicola Brandt, Jan Philipp Willmer, Maurilyn S. Ayon-Olivas, Veronika Banicka, Martin Witt, Andreas Wree, Isabel Groß, Anne Gläser, Jens Hausmann, Anja U. Bräuer

**Affiliations:** 1Research Group Anatomy, Division of Human Medicine, School of Medicine and Health Sciences, Carl von Ossietzky University Oldenburg, 26129 Oldenburg, Germany; 2Institute of Anatomy, Rostock University Medical Center, 18057 Rostock, Germany; 3Research Center for Neurosensory Science, Carl von Ossietzky University Oldenburg, 26129 Oldenburg, Germany

**Keywords:** PRG3/LPPR1, PRG5/LPPR5, PRG4/LPPR2, neuronal outgrowth, hippocampal neurons

## Abstract

The functional importance of neuronal differentiation of the transmembrane proteins’ plasticity-related genes 3 (PRG3) and 5 (PRG5) has been shown. Although their sequence is closely related, they promote different morphological changes in neurons. PRG3 was shown to promote neuritogenesis in primary neurons; PRG5 contributes to spine induction in immature neurons and the regulation of spine density and morphology in mature neurons. Both exhibit intracellularly located C-termini of less than 50 amino acids. Varying C-termini suggested that these domains shape neuronal morphology differently. We generated mutant EGFP-fusion proteins in which the C-termini were either swapped between PRG3 and PRG5, deleted, or fused to another family member, plasticity-related gene 4 (PRG4), that was recently shown to be expressed in different brain regions. We subsequently analyzed the influence of overexpression in immature neurons. Our results point to a critical role of the PRG3 and PRG5 C-termini in shaping early neuronal morphology. However, the results suggest that the C-terminus alone might not be sufficient for promoting the morphological effects induced by PRG3 and PRG5.

## 1. Introduction

Complex sequential processes during prenatal and early postnatal periods are required to initiate periods of outgrowth, leading to neuronal differentiation and subsequent formation of proper neuronal connectivity [1,2]. In the first step of differentiation, neurites extend from the spherical soma of a neuron. One of these neurites will become an axon; the others will become primary dendrites, which are the prerequisite for the development of the dendritic trees which later give rise to spines. Spines enable the neuron to form synaptic contacts required for signal transduction and plasticity [3,4,5,6]. These developmental processes are controlled by multiple extracellular cues, but can also be modulated intracellularly. Several proteins have been identified that control the establishment of neuronal outgrowth [3,5,6]. 

Recently, the role of two transmembrane proteins, plasticity-related gene 3 (PRG3; also referred to as lipid phosphate phosphatase-related gene 1 (LPPR1)), and plasticity-related gene 5 (PRG5/LPPR5) in neuronal differentiation has been shown [7,8,9,10]. PRG3 and PRG5 belong to the family of plasticity-related genes (PRGs), which defines a subclass of the Lipid Phosphate Phosphatase (LPP) superfamily. The PRG family comprises five, vertebrate, brain-specific membrane proteins, which influence lipid phosphate signaling and thereby promote filopodia formation, neurite extension, axonal sprouting and reorganization after lesions [7,8,9,11,12,13,14,15,16,17]. Although PRG3 and PRG5 share the highest structural and sequence similarities within the PRG family, their short (~50 amino acids) intracellularly located C-termini are unique [16,18], which indicates that they might exert different functions in cells compared to each other.

Individual expression patterns during brain development in mice gave rise to the assumption that PRGs exert different regulatory mechanisms in the CNS [16,18,19]. In addition, although the sequence of PRG3 is closely related to that of PRG5, they promote different morphological changes in neurons. PRG3 is involved in neurite outgrowth and promotes neurite shaft protrusion [9], and its expression is changed after the overexcitation of neurons induced by kainic acid [14]. PRG5, on the other hand, binds to phospholipids and is involved in spine formation and synaptic stabilization [8]. Recently, we systematically investigated the expression pattern of PRGs in mouse brain development, which has drawn our attention to another family member, PRG4 (LPPR2) [18]. PRG4 was found to be ubiquitously expressed throughout all developmental stages and all brain areas examined and is predominantly expressed in neurons. Interestingly, PRG4 did not induce filopodia formation in non-neuronal cells. Despite being highly expressed during development, the function of PRG4 is still elusive. 

Functional tests suggested that both PRG3 and PRG5 contribute to an mDia/Cdc42-independent pathway promoting actin-enriched protrusions and neurite growth in cell lines and immature primary neurons [7,15]. The C-terminal domain of PRG3 has been shown to be essential for inducing membrane protrusions [20], as well as increasing neurite outgrowth [10].

PRG5 growth-promoting activity is mediated by the C-terminus and needs to be localized near the plasma membrane [7,8]. Our recent study showed an interaction of PRG5 with lipids, especially phosphatidylinositol phosphate (PIPs), and that this interaction is mediated via the C-terminal domain [8]. Therefore, the C-terminal domain apparently plays a role during neuronal differentiation. The question remains, as to how much the endogenous, unique C-terminus, is involved.

These findings prompted us to investigate the involvement of the varying C-terminal domains of PRG3 and PRG5 in stages of neuronal differentiation during which neurite outgrowth happens (Stage 3, [21]). Therefore, we generated different mutant plasmids in which C-terminal domains were either deleted, swapped between PRG3 and PRG5, or fused to PRG4 and analyzed their influence after overexpression of the respective fusion proteins in immature hippocampal neurons. 

Taken together, recent evidence suggests the importance of the C-terminal domain of PRG3 and PRG5 to exert their specific roles during neuronal differentiation and function. However, how exactly the C-termini are involved, is still unresolved. Thus, we aimed to address the question as to whether the C-terminal domain is solely responsible for the different functions of PRG3 and PRG5, or whether other domains in these proteins are also involved. Therefore, different fusion proteins were generated with exchanges in the different domains between PRG3, PRG4 and PRG5.

## 2. Results

### 2.1. Membrane Expression of PRG3 and PRG5, also with Swapped C-Termini, but Not with PRG3∆C and PRG5∆C

Previous findings [7,8,9] and the different C-terminal domains of PRG3 and PRG5 gave rise to the assumption that these domains shape neuronal morphology differently. Structure models based on the amino acid sequences of both PRG3 and PRG5 are shown in Figure 1A and Figure 2A. To investigate the influence of the endogenous C-terminal domain more precisely, we generated EGFP-fusion proteins in which it was either deleted or swapped between PRG3 and PRG5 (Figure 1B,C and Figure 2B,C). In order to prove that the deleted and swapped C-terminal domain fusion proteins were expressed, we performed western blot analyses (Appendix A). Western blot analyses of protein lysates resulting from overexpression of PRG3, PRG3∆C, PRG3-C5, PRG5, PRG5∆C and PRG5-C3 in cultured HEK293H cells with an anti-GFP antibody confirmed expression of PRG3 and PRG3-C5. Double bands were found at the expected size of ~63 kDa and PRG5 and PRG5-C3 bands on a slightly lower molecular level at ~62.9 kDa, as well as a single band at 26.9 kDa representing GFP alone. The deletion constructs PRG3∆C and PRG5∆C showed bands at ~57 kDa (Appendix A). In addition, and as also recently described by us at least for PRG5, the antibody also detected a band above 150 kDa [19]. As a control, no band can be seen at 63 kDa or 57 kDa in EGFP-transfected control cells (Appendix A). Taken together, all experiments confirmed the expression of our fusion proteins.

Immunocytochemical analysis of non-neuronal cells recombinantly expressing PRG3 and PRG5 revealed that the fusion proteins were localized in the plasma membrane (Figure 1A and Figure 2A). This is an important aspect, as PRGs are only functional when thus localized [7,10,20]. Therefore, the question arose as to which of our generated fusion proteins is localized there. In contrast to PRG3 and PRG5, localization in the plasma membrane could not be observed in both C-terminal deletion fusion proteins (PRG3∆C, PRG5∆C) (Figure 1B and Figure 2B). Moreover, cells overexpressing PRG3 tended to develop lamellipodia-like structures. PRG5, on the other hand, induced a more filopodial morphology. Comparatively, cells overexpressing the C-terminal-truncated versions exhibited almost no filopodial structures (Figure 1A,B and Figure 2A,B). Non-neuronal cells were used to test whether a neuronal protein is capable of altering the morphology of these cells. The generated fusion proteins seem to generally influence filopodia formation, pointing to a more general mechanism.

Our next question was: Do PRG3 and PRG5 need their endogenous C-termini to induce morphological changes? Interestingly, when the fusion proteins with swapped C-terminal domains were overexpressed in non-neuronal cells, these cells tended to restore the morphological phenotype of either PRG3 or PRG5. Both fusion proteins (PRG3-C5, PRG5-C3) appeared to be localized in the plasma membrane and exhibited an increased number of filopodia as compared to cells expressing the C-terminal truncated versions (PRG3∆C, PRG5∆C). While overexpression of PRG3-C5 tended to induce a more lamellipodial phenotype, overexpression of PRG5-C3 resulted in a more filopodial formation (Figure 1C and Figure 2C). 

In addition, we analyzed the fusion proteins in immature primary hippocampal neurons after two days in culture (DIV2) (Figure 3A, Figure 4A and Appendix A). Neurons were identified by immunostaining for tubulin. Likewise, during this early phase of neuronal differentiation, overexpression of full-length PRG3 and PRG5, as well as the swapped C-terminal fusion proteins PRG3-C5 and PRG5-C3, resulted in a strong signal and punctate distribution in the plasma membrane. Neurons overexpressing either one or the other C-terminally truncated versions (PRG3∆C, PRG5∆C) did not show areas in which plasma membrane localization could be observed. PRG3∆C, as well as the PRG5∆C protein, was mainly found in the cytosol. All of the fusion proteins also induced morphological changes in immature neurons (Figure 3 and Figure 4), which we analyzed further. 

### 2.2. PRG3 and PRG5 C-Termini Are Important Players in Early Neuronal Differentiation, No Matter Whether the C-Term Came from One or the Other Protein

These obvious morphological changes prompted us to investigate the influence of PRG3 and PRG5, as well as the consequences of deleting (PRG3∆C and PRG5∆C) and swapping the C-terminal domains (PRG3-C5 and PRG5-C3) on shaping neuronal morphology in early stages of differentiation. Overexpression experiments were started at DIV1, and the analysis was performed at DIV2, representing stage 3 of neuronal differentiation [21]. As PRG3 promotes neurite growth [9], we analyzed the number of neurites, as well as major neurite length, in immature cultured neurons. Branching is a hallmark of early neuronal differentiation, so we questioned whether the C-terminal domains affect neurite branching. In addition, we examined the impact of the six fusion proteins on protrusion length, as protrusions on neurites are essential for neuronal development and are the foundation of dendritic spines. Cell-membrane extensions shorter than 10 µm were classified as protrusions. Individual subgroups consisting of protrusions <2 µm (short protrusions), 2–<5 µm (medium protrusions), and 5–<10 µm (long protrusions) were analyzed.

First, overexpression of PRG3∆C, as compared to PRG3, and therefore deletion of the C-terminus, resulted in significant differences, more specifically decreases, in all parameters studied (Figure 3B–G; for median/IQR see Appendix A and *p* values see Table 1). This suggests that the C-terminal domain of PRG3 is important to promote aspects of neuronal differentiation. However, the overexpression of PRG3-C5 led to changes in all parameters examined that were not significantly different compared to those of PRG3, but were significantly different to the truncated C-terminal version of PRG3 (PRG3∆C) (Figure 3B–G; for median/IQR see Appendix A and *p* values see Table 1). Since PRG3-C5 could restore the function, it is apparently independent of the endogenous C-terminal domain of PRG3.

Only the C-terminally deleted version of PRG5 showed a significant decrease in the number of protrusions of all subgroups, as well as in the major neurite length and the number of branching points (Figure 4C–G; for median/IQR see Appendix A and *p* values see Table 2). This is independent of the endogenous C-terminal domain of PRG5, since PRG5-C3 is capable of restoring the effect (Figure 4C–G; for median/IQR see Appendix A and *p* values see Table 2). 

Interestingly, the number of neurites was neither altered by the truncated C-terminal version of PRG5 (PRG5∆C) compared to full-length PRG5 and PRG5-C3, nor by PRG5-C3. This suggests that the C-terminus does not affect the number of neurites in PRG5 overexpressing neurons. (Figure 4B; for median/IQR see Appendix A and *p* values see Table 2). 

The induction of morphological changes in immature hippocampal neurons apparently relies on the C-terminal domain of PRG3 as well as PRG5. This seems to be independent of their endogenous C-terminal domains, since altering these domains did not lead to changes in neuronal morphology in any parameters analyzed, except for PRG5 regarding the number of neurites.

### 2.3. PRG3 and PRG5 C-Terminal Domains Induce Morphological Changes in PRG4 Overexpressing Non-Neuronal Cells and Neurons—But to a Lesser Extent

Recently, we showed that, unlike PRG3 and PRG5 family members, PRG4 does not induce filopodial outgrowth or any morphological changes in non-neuronal cells [18]. Furthermore, it does not localize to the plasma membrane of filopodia. To investigate more precisely how the C-terminus of PRG3 and PRG5 is involved in changing cellular morphology, we took advantage of the non-inducing activities of PRG4. We generated EGFP-fusion proteins, in which the C-terminal domain of PRG4 was deleted and replaced by either the C-terminal domain of PRG3 (PRG4-C3) or PRG5 (PRG4-C5). Moreover, we addressed the question of whether PRG4 is directed to the plasma membrane by the C-terminal domains of PRG3 and PRG5. A schematic depiction of the respective proteins is shown in Figure 5A–C. If the C-terminus of PRG3 and PRG5 is responsible for the induction of the morphological effects in immature neurons, the phenotype of PRG4-C3- or PRG4-C5-overexpressing cells should be the same. Firstly, the fusion proteins were validated in non-neuronal cells by performing a western blot analysis. We found double bands at the expected molecular weight of ~63 kDa (Appendix A). Interestingly, no band above 150 kDa could be seen in protein lysates from PRG4-C5 overexpressing cells.

Indeed, unlike PRG3 or PRG5, the overexpression of PRG4 in non-neuronal cells led to a similar phenotype as previously described in [18], with a mainly intracellular localization of the fusion protein (FLAG-tagged) and the inability to induce the formation of membrane protrusions (Figure 5A). In contrast, PRG4-C3- and PRG4-C5-overexpressing cells clearly showed increased sheet-like structures and membrane protrusions that co-localized with F-actin structures. The intracellular distribution of PRG4-C3 and PRG4-C5 showed a punctuated pattern, similar to PRG3 and PRG5, and both were localized in the plasma membrane (Figure 5B,C).

Since overexpression of PRG4-C3 and PRG4-C5 in non-neuronal cells resulted in changes in morphology, the question was: is the C-terminal domain of PRG3 and PRG5 sufficient to induce a comparable phenotype in immature cultured neurons?

Similar to our previous analyses (Figure 3 and Figure 4), immature neurons overexpressing PRG4, PRG4-C3 and PRG4-C5 all displayed protrusions along the neurites (Figure 6, Figure 7 and Appendix A). Despite this, when overexpressing PRG4, PRG3, or PRG4-C3, no significant differences were observed, neither in the number of neurites, nor in the major neurite length or the number of branching points (Figure 6B–D; for median/IQR see Appendix A and *p* values see Table 3). A considerable and significant increase was observed in all protrusions of the different subgroups of PRG3 as compared to PRG4 (Figure 6E–G; for median/IQR see Appendix A and *p* values see Table 3). Interestingly, the overexpression of PRG4-C3 only showed an increased tendency in the number of short protrusions, which is in-between PRG3 and PRG4, as there are neither statistical significances between PRG3 and PRG4-C3, nor between PRG4 and PRG4-C3 (Figure 6E; for median/IQR see Appendix A and *p* values see Table 3). Since the morphological phenotype is not the same, this points to a functional role of additional domains outside the C-terminal domain of PRG3, which are not present in PRG4. 

To shed light on what kind of morphological changes are induced by the C-terminal domain of PRG5, we performed measurements as previously. As compared to PRG4, overexpression of PRG5 induced higher numbers of all morphological parameters examined (Figure 7B–G; for median/IQR see Appendix A and *p* values see Table 4). In none of the parameters studied, with the exception of short protrusions, was PRG4-C5 capable of mimicking the effect of PRG5. In the case of short protrusions, neurons overexpressing PRG4-C5 showed a significant increase as compared to PRG4, suggesting an involvement of the PRG5 C-terminus. However, the function was not completely restored, as the number was still decreased as compared to PRG5 (Figure 7E; for median/IQR see Appendix A and *p* values see Table 4).

To summarize the impact on neurite branching and outgrowth as well as membrane protrusions: as compared to PRG4, PRG5 induced higher numbers in every single parameter investigated, with an unequivocally strong effect on short protrusions. Moreover, in this subgroup of short protrusions, the C-terminal domain partially restored the effect, but not to the full extent. In all other conditions, the C-terminus was not sufficient to mimic the effects of the full-length protein. This indicates that other domains play a role in mediating the neuronal differentiation processes. Which domains are responsible still need to be clarified.

## 3. Discussion

Our study showed that PRG3 and PRG5, unlike PRG4, are involved in the induction of protrusions and neurite outgrowth. What became clear in this study is that while the C-terminal domains of PRG3 and PRG5 are important for the induction of protrusions, this is independent of which of the two endogenous C-terminal domains are fused to the respective core protein. The C-terminal domains of PRG3 and PRG5 are interchangeable. Since the C-terminal domain is located intracellularly, it can act as a mediator initiating intracellular signaling pathways. Recently published work corroborated the importance of the C-terminal domain of PRG3 in forming filopodia and in the elongation of neurites [10,15]. A direct interaction of PRG3 and the Ras GTPase exchange factor RasGEF1 has been shown, which depends on the C-terminal domain [10,22]. Furthermore, in cortical neurons, this interaction induced filopodia formation and neuronal plasticity via downstream effectors such as Raf and MEK [10]. Additionally, deletion of the C-terminal domains of PRG3 and PRG5 diminished the effects promoted by full-length PRG3 and PRG5 under almost all conditions examined, corroborating the findings of other groups [7,10,15,20]. Possible reasons for this could be a failed plasma-membrane integration via an impaired intracellular transport of the protein, as well as an interruption of intracellular signaling pathways. 

The most salient finding, however, was that PRG3 and PRG5 C-terminal domains fused to the respective other PRG, clearly promoting growth and branching, but this is not the case when fused to PRG4, except for major neurite length. This suggests that other domains not present in PRG4 are also important for this function. Sequence alignment in our recent study revealed that the largest differences between PRG3, -4, and -5 are in the intracellular N- and C-termini, the first intracellular loop, and the second extracellular loop [18]. There is, however, sequence similarity between PRG3 and PRG5, for instance in the intracellular loops, which does not occur in this form in PRG4. Which additional domains play a role in neuronal differentiation thus remains to be clarified. 

### 3.1. Plasma Membrane Localization Is Essential for PRG3 and PRG5 Function

The C-terminal domain of PRG3 and PRG5 is essential for plasma-membrane localization and function [7,10,20]. This is reflected in our experiments showing that the truncated versions are not localized in the plasma membrane, whereas the swapped fusion proteins are. Previous studies revealed a membrane localization after overexpression of membrane-targeted C-terminal domains of PRG3 and PRG5 [7,10] alone, as well as solely expressed C-terminal domain of PRG3 [20]. PRG4 was observed to be mainly localized intracellularly in non-neuronal cells [18]. Overexpression of PRG4-C3 and PRG4-C5 showed membrane localization, and in non-neuronal cells it is able to induce phenotypes similar to PRG3 and PRG5 overexpression. Thus, a possible explanation could be that when fused to PRG4, the C-terminal domain of PRG3 and PRG5 lead to a shift towards the plasma membrane. Other studies reported that overexpression of the membrane-targeted C-terminus of PRG3 or PRG5 alone increased filopodia and neurite growth in N1E-115 cells and non-neuronal cells [7,10]. This is in line with our findings, since PRG4-C3 and PRG4-C5 both increased filopodia-like structures in non-neuronal cells. In immature primary hippocampal neurons, on the other hand, the situation was different. Both fusion proteins, PRG4-C3 and PRG4-C5, although they localized to the plasma membrane, were barely able to induce the effects of PRG3 and PRG5 overexpression. This suggests that the C-terminal domain is not capable of inducing these effects alone. Our results indicate that the induction of complex neuronal morphology requires other, additional, domains that are not present in PRG4. Another possibility would be that the core part of PRG4 has rather an inhibitory effect.

### 3.2. The C-Terminus of PRG3 and PRG5 Plays a Significant Role in Major Neurite Growth

During development, a major neurite typically becomes the axon [23]. Major neurite lengths of PRG3-, as well as PRG5-overexpressing neurons, are affected by truncation of the C-terminal domain. Interestingly, it has been shown that the PRG3 C-terminus is essential for axon outgrowth, given that it is localized near the plasma membrane [10]. A growth-promoting effect of PRG5 C-terminus in axon elongation in early cortical neurons has also been shown [7]. This is corroborated by our experiments but is extended by the fact that the C-terminal domains of PRG3 and PRG5 are interchangeable. Interestingly, this growth-promoting effect on the major neurite is also seen in PRG4-C3 and PRG4-C5. Recently, functional studies revealed PRG3 as a strong player influencing axonal growth in vitro, in vivo and following spinal-cord lesions, pointing to a regenerative function [10,24]. 

### 3.3. Only the Full-Length PRG3 or PRG5 Protein Enhances Neurite Branching 

In addition, to enhance neurite outgrowth, both PRG3 and PRG5 contribute to neurite branching, and to promote branching, the C-terminal domains of either PRG3 or PRG5 are involved. This is in line with the study by [10] showing an exclusive role of the PRG3 C-terminal domain. However, we show here that both PRG3, at least tendentially, and PRG5, have a larger capacity than PRG4 to promote branching in immature primary hippocampal neurons, which was not seen with PRG4-C3 and PRG4-C5. Therefore, one can conclude that the branching induced by PRG3 and PRG5 needs the C-terminal domain in combination with additional domains of PRG3 or PRG5 core protein that are not present in PRG4. 

### 3.4. The C-Terminus of PRG3 or PRG5 Is Insufficient to Induce Membrane Protrusions 

Our study clearly shows that particularly the C-terminal domain of PRG3 and PRG5 plays an important role in mediating the formation of membrane protrusions. This is in line with several recent studies [7,8,9,10,20]. For example, previous results showed that the knock down of PRG3 impedes the capacity of immature neurons at DIV4 to initiate protrusions of 2 to 5 µm length [9]. After co-expression, PRG3 and PRG5 were shown to be localized in plasma membrane protrusions in neuronal cells [20] and increased the generation of protrusions beyond that of a single expressed PRG3 or PRG5. The authors hypothesized that they may act in concert during neuronal differentiation. Therefore, it is of functional importance to clarify which domains are essential to induce membrane protrusions in these particular family members. Understanding the physiological roles of PRGs in neurons can later help to elucidate pathological conditions. A better understanding of the functional domains of PRGs could lead to new strategies of regulation of the protein and, consequently, the development and stabilization of spines.

Interestingly, the C-terminal domain of PRG3 and PRG5 alone is sufficient to promote the induction of protrusions, as long as this is fused to either PRG3 or PRG5, but not to PRG4. This result indicates that the C-termini of PRG3 and PRG5 are important, but cannot induce protrusions without the core protein of PRG3 and PRG5. Thus, this morphological change is also dependent on an additional domain in the PRG3 and PRG5 core protein, which is not present in PRG4.

## 4. Material and Methods

### 4.1. Animals

Timed-pregnant mice (C57BL/6J) were obtained from the central animal facility of the Carl von Ossietzky University Oldenburg and kept under standard laboratory conditions (12 h light/dark cycle; 55 ± 15% humidity; 24 ± 2 °C room temperature (RT)). Water and food were available ad libitum. All experiments were carried out in accordance with the institutional guidelines for animal welfare and approved by the “Niedersächsisches Landesamt für Verbraucherschutz und Lebensmittelsicherheit” (33.19-42502-04-18/2766). Embryonic day 18/19 (E18/19) mice were used for culturing primary hippocampal neurons. The day of the vaginal plug was designated E0.5 following mating.

### 4.2. Plasmids

The following expression plasmids were used (Table 5): 

For plasmid cloning, pEGFP-N1-mPRG3, pEGFP-N1-rPRG5 and pFLAG-CMV2-mPRG4 were used as origins to create the PRG3ΔC, PRG3-C5, PRG5ΔC, PRG5-C3, PRG4-C3 and PRG4-C5 plasmids, respectively (Figure 1, Figure 2 and Figure 5). Deletion of the intracellularly located C-terminal encoding domains (mPRG3ΔC: 48 amino acids deleted, rPRG5ΔC: 42 amino acids deleted) was achieved by PCR amplification of the residual upstream located sequence using full-length pEGFP-N1-mPRG3 and pEGFP-N1-rPRG5 as template DNA. Oligonucleotides used were synthesized by Eurofins genomics (Ebersberg, Germany) and can be found in Appendix A. Primer pairs were designed to introduce restriction sites to the 3′-ends and 5′-ends of the amplified DNA fragments (forward primers were designed to bind upstream of the gene’s open-reading frames to include upstream restriction sites that were already included in the template plasmids sequence; a *Bam*HI restriction site was added to the 5′-end of the reverse primer). PCR products were purified by agarose gel electrophoresis and a QIAEX II Gel extraction kit (Qiagen, Hilden, Germany). Both plasmids were cloned via digestion with XhoI and BamHI (both from Thermo Fisher Scientific, Waltham, MA, USA). 

For generating PRG3-C5 and PRG5-C3 plasmids, the C-terminal domains (including transmembrane domain 6) of PRG3 and PRG5 were swapped. Exchange of the C-terminal domain was performed by digesting the vectors with AgeI and BmgBI (both from Thermo Fisher Scientific) according to the manufacturer’s instructions and inserting the resulting C-terminal fragments into the other plasmid.

The PRG4-C3 plasmid was generated from full-length pFLAG-CMV2-mPRG4 and pEGFP-N1-mPRG3. An additional XhoI restriction site upstream of the start codon in the PRG4 sequence was introduced by PCR. For primer sequences see Appendix A. PCR products were purified as described above. After digestion and purification of pEGFP-N1-mPRG3 and PCR fragments with XhoI and BmgBI, PRG4 without a C-terminus (insert) was ligated to the remaining C-terminus of PRG3 in the pEGFP-N1-mPRG3 vector. The PRG4-C5 plasmid was generated from full-length pFLAG-CMV2-mPRG4 and pEGFP-N1-rPRG5. Vectors were digested with BmgBI and HindIII (Thermo Fisher Scientific) to obtain a PRG4 without a C-terminus (insert) which was subsequently ligated to the remaining C-terminus of PRG5 in the pEGFP-N1-rPRG5 vector.

The resulting plasmids were verified by gel electrophoresis using HDGreen Plus DNA Stain (Intas Science Imaging Instruments, Göttingen, Germany) as a marker and confirmed by sequence analysis using CLC sequence viewer 8.0 software (Qiagen, Hilden, Germany).

### 4.3. Primary Mouse Hippocampal Neuron Cultures 

Hippocampal primary neurons were prepared from E18/19 mouse embryos [25]. Hippocampi of all embryos from one pregnant mouse were collected, pooled and washed twice in HBSS (Hank’s Buffered Salt Solution, Thermo Fisher Scientific). The tissue was incubated in 5 mL HBSS and 500 µL 2.5% trypsin (Life Technologies Limited, Paisley, UK) for 10–15 min at 37 °C, re-suspended in 3 mL MEM plating medium (Life Technologies Limited, Paisley, UK) supplemented with 10% horse serum (Thermo Fisher Scientific), 0.6% glucose (Sigma-Aldrich Chemie, Steinheim, Germany), and 100 U/mL penicillin with 100 µg/mL streptomycin (PAN-Biotech, Aidenbach, Germany). After dissociation, neurons were plated in a plating medium onto poly-L-lysine coated coverslips (0.2 mg/mL; Sigma-Aldrich) at a density of 1 × 10^5^ cells/well of a 24-well plate. Three hours after plating, cells were washed twice with 1× PBS (Phosphate-buffered Saline, Life Technologies Limited), incubated and routinely maintained in Neurobasal A medium, supplemented with 2% B27, 0.25% glutamine (all from Life Technologies Limited) and 100 U/mL penicillin with 100 µg/mL streptomycin at 37 °C and 5% CO_2_. 

### 4.4. Cell Culture and Transfection 

Primary neurons were transfected at day in vitro (DIV) 1.5 with the respective plasmids PRG3, PRG3ΔC, PRG3-C5, PRG5, PRG5ΔC, PRG5-C3, PRG4, PRG4-C3, PRG4-C5 with Effectene (Qiagen, Hilden, Germany) according to the manufacturer’s instructions. 

Human embryonic kidney (HEK293 (Thermo Fisher Scientific) /HEK293H (ATCC CRL-1573, LGS Standards GmbH, Wesel, Germany)) cells were routinely maintained at 37 °C in humidified 5% CO_2_-enriched atmosphere in DMEM (PAN-Biotech) supplemented with 10% FCS, 1% L-glutamine (Life Technologies Limited) and 100 U/mL penicillin with 100 µg/mL streptomycin (PAN-Biotech). 

For transfection experiments, 5 × 10^4^ cells/well of a 24-well plate were plated onto coverslips. After ~24 h, the cells were transfected with 0.5 µg of the respective plasmids by calcium phosphate transfection. All plasmids encode for GFP fusion proteins; for abbreviations see Table 5.

For validation of plasmid expression by western blot, transfection experiments were conducted with 1 × 10^6^ HEK 293H cells per 100-mm Petri dish. Twenty-four hours after plating, cells were transfected with 22 µg of the plasmids by calcium phosphate transfection. The pEGFP-N1 plasmid was used as a control. 

### 4.5. Western Blot Analysis

HEK293H cell lysates were used for western blot analysis (Appendix A). Nineteen to 24 h after transfection, HEK293H cell lysates were prepared to obtain recombinant protein using the µMACS GFP Isolation Kit (Miltenyi Biotec, Bergisch Gladbach, Germany) according to the manufacturer’s instructions. Lysates were analyzed by performing an SDS-PAGE with 10% acrylamide under reducing conditions, with a subsequent western blot. Blocking was performed with 5% non-fat dry milk-powder solution (in 1× PBS or 1× TBST) for 1 h at room temperature (RT). Proteins were analyzed using the following antibody (Appendix A): mouse anti-GFP ((JL-8) 632380, Clontech/Takara Holdings, Kyoto, Japan) (1:2500). Incubation was overnight at 4 °C. Washing was three times with 1× PBS containing 0.1% T (Tween 20; Carl Roth, Karlsruhe, Germany) for 10 min. The secondary antibody used was sheep anti-mouse-HRP (conjugated to horseradish peroxidase) (GENA931-100UL, Sigma-Aldrich Chemie) (1:10,000) (Appendix A). Specimens were incubated for 1.5 h at RT and washed three times with PBS/0.1% T for 10 min. Proteins were detected with a ChemiDoc MP Imaging System (Bio-Rad Laboratories, Hercules, California, USA) using ECL Plus Western Blotting Substrate (Thermo Fisher Scientific) or Clarity Max Western ECL Substrate Kit (Bio-Rad Laboratories). All Blue Precision Protein Plus (Bio-Rad Laboratories) was used as a protein standard.

### 4.6. Immunocytochemistry

Primary hippocampal neurons were washed once with 1× PBS and fixed at DIV2 (16–17 h after transfection) in ice-cold 4% paraformaldehyde (Merck KGaA, Darmstadt, Germany) with 15% sucrose in 1× PBS for 10 min at RT. HEK293H cells were washed once with 1× PBS and fixed for 16–17 h after transfection for 20 min. Subsequently, they were washed three times with 1× PBS for 10 min and permeabilized in 0.1% Triton X-100/PBS + 0.1% sodium citrate for 3 min. Cells were then washed with 1× PBS and blocked in 10% FCS/1% NGS/PBS at RT for 1 h. For immunostaining, the following antibodies were applied in 5% FCS/1% NGS/PBS overnight at 4 °C: mouse-anti-GFP (JL-8) (1:1000), mouse-anti-FLAG (HM-2) (1:1000) and chicken-anti-TUBBIII (1:1000) (Appendix A). After washing three times with 1× PBS for 10 min, primary antibodies were detected with the following secondary antibodies (incubation 90 min at RT): goat-anti-mouse Alexa Fluor 488 (1:1000), and rabbit-anti-chicken TRITC (1:400) or goat-anti-chicken Alexa Fluor 568 (1:1500) (Appendix A). Phalloidin-iFluor 555 and DAPI (counterstaining for Nuclei) were incubated together with secondary antibodies. Coverslips were again washed three times for 10 min with 1× PBS and cells were mounted with Immu Mount Vectashield Hard Set Mounting medium (Vector Laboratories, Burlingame, CA, USA).

### 4.7. Image Acquisition and Morphological Analysis

For morphological analysis, fluorescence images of primary hippocampal neurons and HEK293H cells were captured with an IX83 inverted imaging system equipped with a DP80 camera (Olympus, Shinjuku, Tokio, Japan) using cellSens Dimension 1.18 (Build 16686) software from Olympus. Images were taken either by using a UPlanSApo 60×/1.35 or a UPlanSApo 100×/1.4 oil objective. The adjustment of brightness, contrast and quantification was performed using cellSens Dimension software version 1.18 (Olympus). Representative confocal co-localization z-stack images were acquired using the Leica TCS SP8 confocal microscope (Leica, Wetzlar, Germany) with the 488-nm line of an argon-ion laser and 552-line of a helium laser. Scanning was performed using an HC PL APO CS2 63×/1.4 or HC PL APO CS2 40×/1.3 oil objective. Z-stacks were sequentially acquired and images are presented as maximum projections of z-stacks. Images were processed using either Leica confocal software (Leica Microsystems, Germany), Image J (NIH, Bethesda, MD), Photoshop CC2020 (Adobe, San Jose, CA, USA), or CorelDraw 2021 and 2017. 

All measurements were performed using 3 independent transfection experiments (*n* = 3), from which neurons were randomly selected for each experimental group (*n* = 30–44 neurons; see Appendix A). More specifically, for each independent experiment, hippocampi from all embryos (7–9 on average) of a pregnant mouse were collected and pooled. For each fusion protein, at least two to three coverslips were imaged per independent experiment to achieve higher variability, resulting in a total of *n* = 10–15 neurons. The polyline tool of the cellSens Dimensions software (Olympus) was used to trace the neurons. Protrusions were manually counted and their length was measured from the tip to the intersection with the protrusion shaft. Protrusions > 10 µm were classified as “neurites”. When originating from another neurite, they were classified as “branching neurites”. The longest neurite arising from the soma was classified as the “major neurite”. The number of protrusions from 0.2 µm to <2 µm, between 2 and <5 µm, and between 5 and <10 µm were quantified as separate subgroups. The mean total process length per neuron was evaluated by summing the lengths of all processes.

### 4.8. Statistics and Data Analysis

Data analysis was performed using GraphPad Prism7 (GraphPad Software Inc., La Jolla, CA, USA) and IBM SPSS Statistics version 26.0.0.0 for Windows (IBM; Armonk, NY, USA). All data are presented as median/interquartile range (IQR) (see Appendix A). Experimental sample sizes were chosen according to the norms of the field. Values were analyzed for normal distribution using the Shapiro–Wilk test for normality with an α-value of 0.05. Since the data were not normally distributed, the datasets were tested for statistical significance between samples greater than two using the Kruskal–Wallis H test and Dunn’s post hoc test. A level of confidence of *p* ≤ 0.05 was adopted (* *p* ≤ 0.05, ** *p* ≤ 0.01, *** *p* ≤ 0.001, n.s. = not significant (*p* > 0.05) (see Table 1, Table 2, Table 3 and Table 4).

## 5. Conclusions

In summary, our results suggest that both PRG3 and PRG5 play a pivotal role in neuronal differentiation; particularly the C-terminal domains of both proteins are important for growth. This is extended by the fact that other, yet unidentified, domains of PRG3 or PRG5 not located in the C-terminal domains are essential for the establishment of certain neuronal differentiation steps. Furthermore, the C-terminal domains are exchangeable for the induction of membrane growth and protrusions, and are also important for plasma-membrane localization. In contrast, the initiation of protrusions is only weakly dependent on PRG4, as it is not capable of inducing membrane effects to the same extent as PRG3 and PRG5. How the action of PRG3 and PRG5 in shaping neurons is achieved still needs further clarification. 

## Figures and Tables

**Figure 1 ijms-23-13007-f001:**
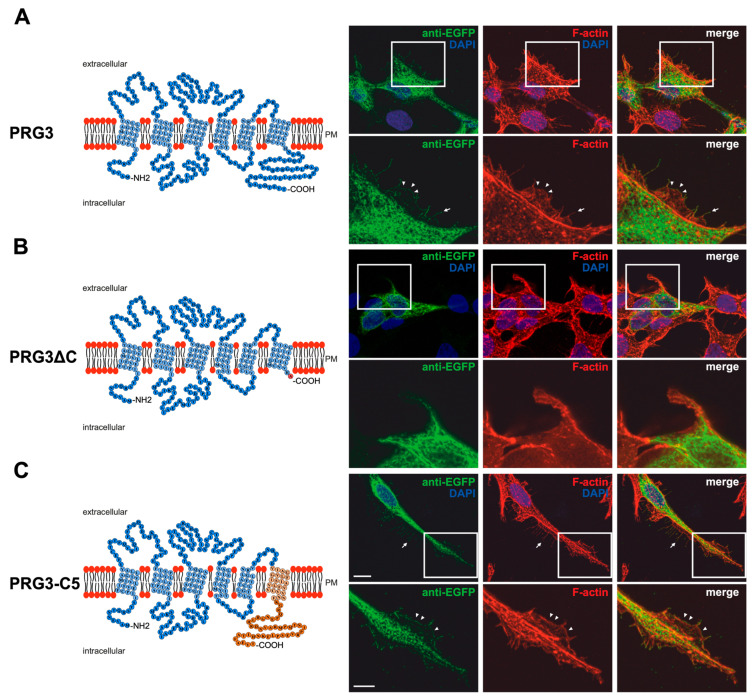
PRG3 and PRG5 C-terminal domains are essential for inducing morphological changes in non-neuronal cells (**A**) Left panel: Schematic depiction of PRG3. Transmembrane domains TM1-TM6 are highlighted in light blue and the C- and N-terminal regions are located on the intracellular side. Single circles represent amino acids (aa) (one letter code). Right upper panel: Confocal stacks of representative HEK293H cells transfected with PRG3-EGFP fusion-encoding plasmids and stained for GFP. Arrows point to filopodia identified by F-actin staining (red, phalloidin). Arrowheads denote areas showing membrane localization. This refers also to (C). PM = plasma membrane. Right lower panel: Higher magnification of representative HEK293H cells transfected as indicated above and shown by white rectangles. (**B**) Left panel: Schematic depiction of PRG3∆C (43 aa deleted). Upper right panel: Confocal stacks of representative HEK293H cells transfected with PRG3∆C-encoding plasmids and stained for GFP (green fluorescent protein). Lower right panel: Higher magnification of representative HEK293H cells transfected as indicated above and shown by white rectangles. (**C**) Left panel: Schematic depiction of PRG3-C5. The sequence of exchanged C-terminal domain residues of PRG5 is shown in orange (aa 252–316). Upper right panel: Confocal stacks of representative HEK293H cells transfected with PRG3-C5-encoding plasmids and stained for GFP. Lower right panel: Higher magnification of representative HEK293H cells transfected as above and indicated by white rectangles. Cell nuclei were stained with DAPI (blue). The scale bar represents 10 µm in the upper panels and 5 µm in the lower panels.

**Figure 2 ijms-23-13007-f002:**
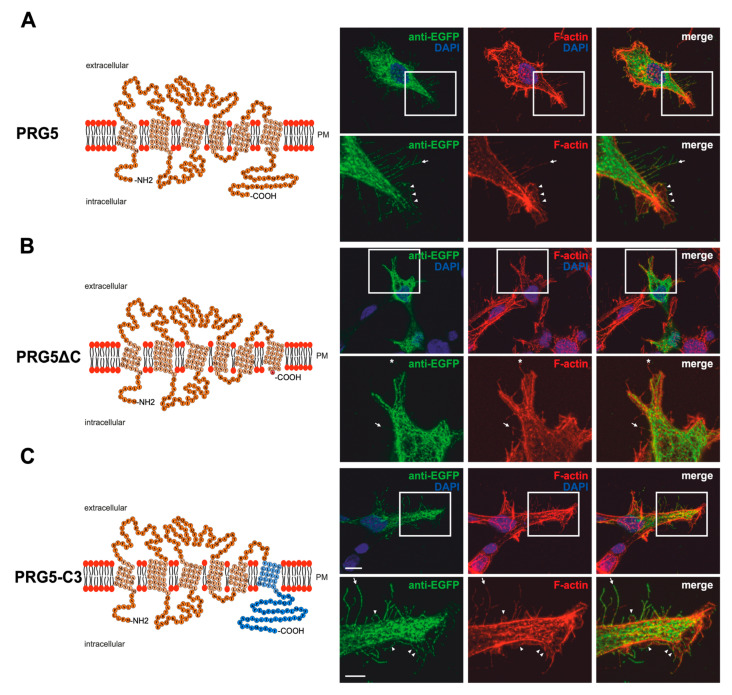
PRG5 and PRG3 C-terminal domains are essential for inducing morphological changes in non-neuronal cells (**A**) Left panel: Schematic depiction of PRG5. Transmembrane domains TM1-TM6 are highlighted in light orange and the C- and N-terminal regions are located on the intracellular side. Single circles represent aa (one letter code). Right upper panel: Confocal stacks of representative HEK293H cells transfected with PRG5-encoding plasmids and stained for GFP. Arrows point to filopodia identified by F-actin staining, (red, phalloidin), while arrowheads denote areas showing membrane localization. This refers also to (**B** and **C**). PM = plasma membrane. Lower right panel: Higher magnification of representative HEK293H cells transfected as above and indicated by white rectangles. (**B**) Left panel: Schematic depiction of PRG5∆C (42 aa deleted). Upper right panel: Confocal stacks of representative HEK293H cells transfected with PRG5∆C-encoding plasmids and stained for GFP. PRG5∆C did not locate all filopodia as denoted by asterisks in the lower panel. Lower right panel: Higher magnification of representative HEK293H cells transfected as above and indicated by white rectangles. (**C**) Left panel: Schematic depiction of PRG5-C3. The sequence of exchanged C-terminal domain residues of PRG3 is shown in blue (aa 260–325). Upper right panel: Confocal stacks of representative HEK293H cells transfected with PRG5-C3-encoding plasmids and stained for GFP. Lower right panel: Higher magnification of representative HEK293H cells transfected as above and indicated by white rectangles. Cell nuclei were stained with DAPI (blue). The scale bar represents 10 µm in the upper panels and 5 µm in the lower panels.

**Figure 3 ijms-23-13007-f003:**
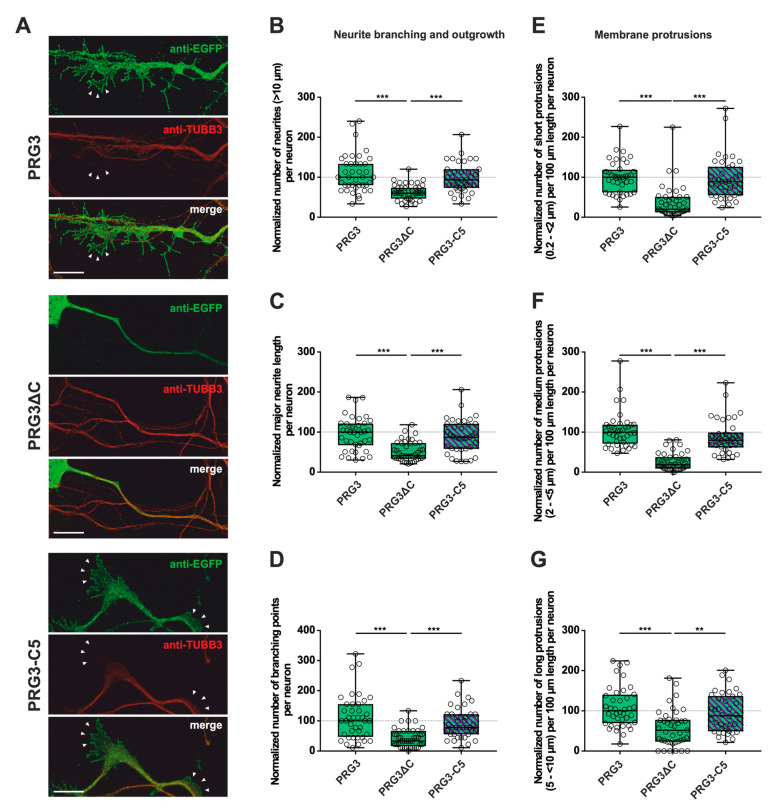
The PRG3 C-terminal domain is an important player in the early development of hippocampal neurons. Subcellular distribution of PRG3-, PRG3∆C- and PRG3-C5-encoding fusion proteins in immature primary hippocampal neurons in culture. Hippocampal neurons in culture were transfected at DIV1 with (**A**) either PRG3- (upper panel), PRG3∆C- (middle panel), or PRG3-C5-encoding plasmids (lower panel), and analyzed at DIV2. Confocal stacks of representative neurons with immunocytochemistry of the respective fusion proteins and stained for GFP to visualize the morphology of the cell or tubulin are shown. Arrowheads denote membrane localization. The scale bar represents 10 µm. Boxplots showing the quantification of (**B**) the total number of neurites (*** *p* ≤ 0.0001); (**C**) the major neurite length (*** *p* ≤ 0.0001); (**D**) the number of branching points (*** *p* ≤ 0.0001) in PRG3-, PRG3∆C- and PRG3-C5-overexpressing neurons. Quantification of membrane protrusions in subgroups based on the length of the (**E**) group of short protrusions (0.2–<2 µm), (**F**) medium protrusions (2–<5 µm) and (**G**) long protrusions (5–<10 µm) (*** *p* ≤ 0.0001, ** *p* ≤ 0.01) in the same neurons (PRG3 *n* = 40, PRG3∆C *n* = 41, PRG3-C5 *n* = 38 neurons). All data are presented as median and IQR. Boxes range from 1st to 3rd quartile. Horizontal lines represent medians. Whiskers represent minimum and maximum. Dots represent the values of individual neurons. Statistical analysis was performed using the Kruskal–Wallis test. A level of confidence of *p* ≤ 0.05 was adopted ** *p* ≤ 0.01, *** *p* ≤ 0.001). PRG3 was always set to 100 and the ratios of changes of the different analyses of morphology are shown (**B**–**G**).

**Figure 4 ijms-23-13007-f004:**
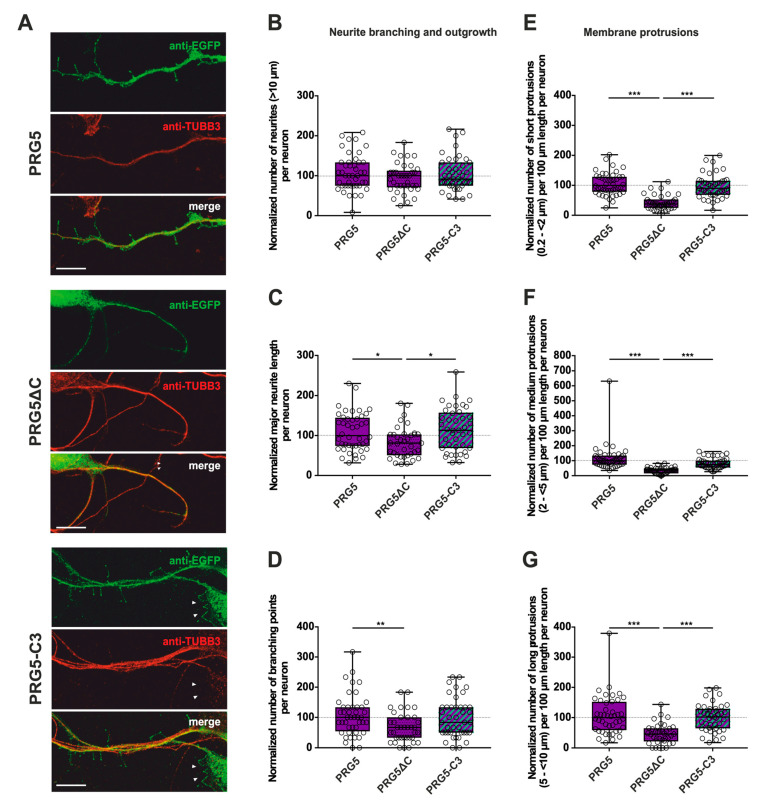
The PRG5 C-terminal domain is an important player in the early development of hippocampal neurons. Subcellular distribution of PRG5-, PRG5∆C- and PRG5-C3-encoding fusion proteins in immature, primary hippocampal neurons in culture. Hippocampal neurons in culture were transfected at DIV1 with (**A**) either PRG5- (upper panel), PRG5∆C- (middle panel), or PRG5-C3-encoding plasmids (lower panel), and analyzed at DIV2. Confocal stacks of representative neurons with immunocytochemistry of the respective fusion proteins stained for GFP to visualize the morphology of the cell or tubulin are shown. Arrowheads denote areas showing membrane localization. The scale bar represents 10 µm. Boxplots showing the quantification of (**B**) the total number of neurites; (**C**) the major neurite length (* *p* ≤ 0.01); (**D**) the number of branching points (** *p* ≤ 0.01) in PRG5-, PRG5∆C- and PRG5-C3-overexpressing neurons. Quantification of membrane protrusions in subgroups based on length in the (**E**) group of short protrusions (0.2–<2 µm), (**F**) medium protrusions (2–<5 µm) and (**G**) long protrusions (5–<10 µm). (PRG5 *n* = 44, PRG5∆C *n* = 37, PRG5-C3 *n* = 41 neurons; *** *p* ≤ 0.0001). All data are presented as median and IQR. Boxes range from 1st to 3rd quartile. Horizontal lines represent medians. Whiskers represent minimum and maximum. Dots represent the values of individual neurons. Statistical analysis was performed using the Kruskal–Wallis test. A level of confidence of *p* ≤ 0.05 was adopted (* *p* ≤ 0.05, ** *p* ≤ 0.01, *** *p* ≤ 0.001). PRG5 was always set to 100 and the ratios of changes of the different morphology analyses are shown (**B**–**G**).

**Figure 5 ijms-23-13007-f005:**
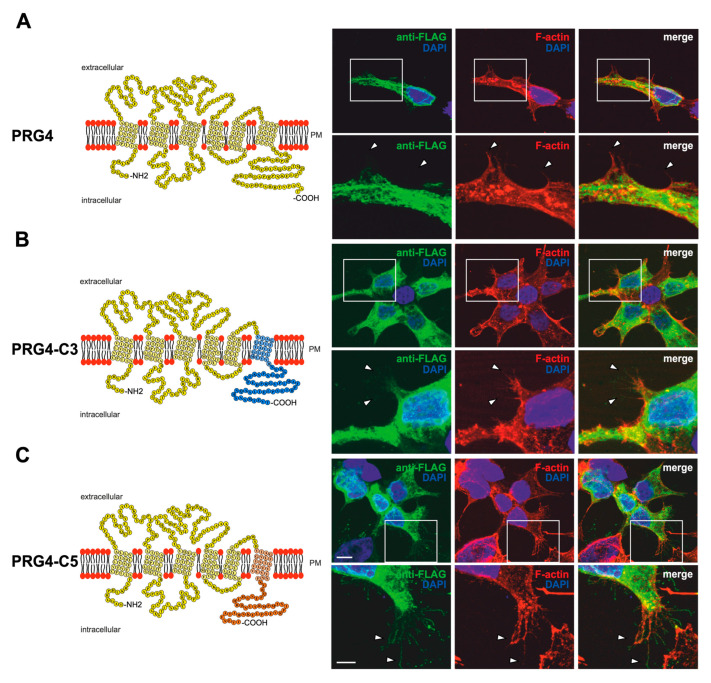
PRG3 and PRG5 C-terminal domains induce morphological changes when fused to PRG4 in non-neuronal cells. (**A**) Left panel: Schematic depiction of PRG4. Transmembrane domains TM1-TM6 are highlighted in light yellow, and the C- and N-terminal regions are located on the intracellular side. Single circles represent aa (one letter code). Upper right panel: Confocal stacks of representative HEK293H cells transfected with PRG4-encoding plasmids and stained for FLAG. Arrowheads point to filopodia identified by F-actin staining (red, phalloidin). This refers also to (**B**) and (**C**)**.** PM = plasma membrane. Lower right panel: Higher magnification of representative HEK293H cells transfected as above and indicated by white rectangles. (**B**) Left panel: Schematic depiction of PRG4-C3. The sequence of exchanged C-terminal domain residues of PRG3 is shown in blue (aa 257–325; including TM6). Right upper panel: Confocal stacks of representative HEK293H cells transfected with PRG4-C3-encoding plasmids and stained for GFP. Lower right panel: Higher magnification of representative HEK293H cells transfected as above and indicated by white rectangles. (**C**) Left panel: Schematic depiction of PRG4-C5. The sequence of exchanged C-terminal domain residues of PRG5 is shown in orange (aa 252–316; including TM6). Upper right panel: Confocal stacks of representative HEK293H cells transfected with PRG4-C5-encoding plasmids and stained for GFP. Lower right panel: Higher magnification of representative HEK293H cells transfected as above and indicated by white rectangles. Cell nuclei were stained with DAPI (blue). The scale bar represents 10 µm in the upper panels and 5 µm in the lower panels.

**Figure 6 ijms-23-13007-f006:**
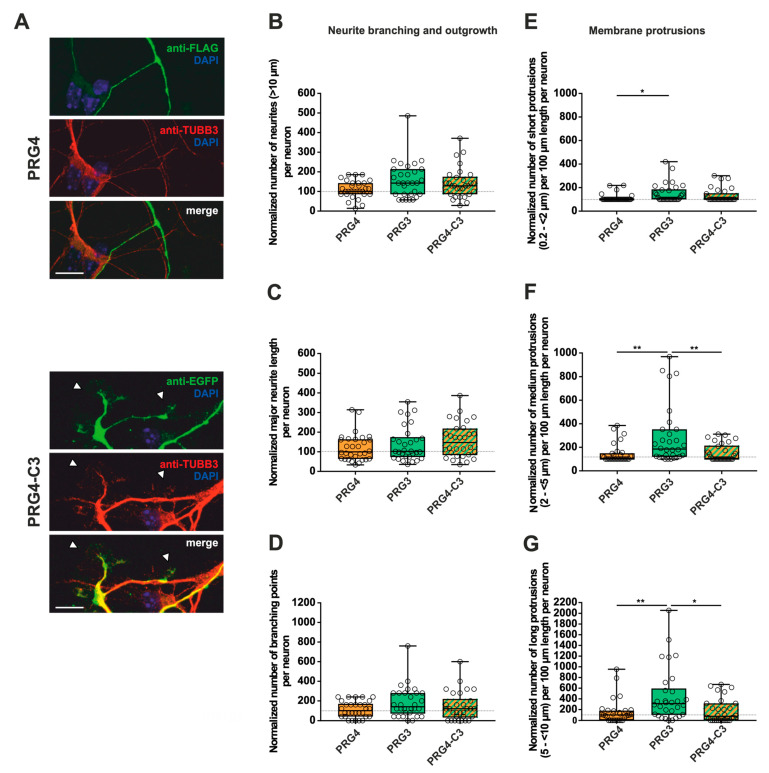
The PRG3 C-terminal domain is crucial for neurite growth and membrane protrusions, but is not capable of restoring the full function of PRG3 when fused to PRG4. Subcellular distribution of PRG4- and PRG4-C3-encoding fusion proteins in immature primary hippocampal neurons in culture. Hippocampal neurons in culture were transfected at DIV1 with (**A**) either PRG4- (upper panel) or PRG4-C3-encoding plasmids (lower panel) and analyzed at DIV2. Confocal stacks of representative neurons with immunocytochemistry of the respective fusion proteins stained for FLAG (green, PRG4) or GFP (green, PRG4-C3) to visualize the morphology of the cell or tubulin are shown. Arrowheads denote membrane localization. The scale bar represents 10 µm. Boxplot showing the quantification of (**B**) the total number of neurites; (**C**) the quantification of major neurite length and (**D**) the number of branching points in PRG4-, PRG3- and PRG4-C3-overexpressing neurons. Quantification of membrane protrusions in subgroups based on length in the (**E**) group of short protrusions (0.2–<2 µm) (* *p* ≤ 0.01) and in subgroups of (**F**) medium protrusions (2–<5 µm) and (**G**) long protrusions (5–<10 µm) (* *p* ≤ 0.01, ** *p* ≤ 0.001). (PRG4 *n* = 30, PRG3 *n* = 30, PRG4-C3 *n* = 30 neurons). All data are presented as median and IQR. Boxes range from 1st to 3rd quartile. Horizontal lines represent medians. Whiskers represent minimum and maximum. Dots represent the values of individual neurons. Statistical analysis was performed using the Kruskal–Wallis test. A level of confidence of *p* ≤ 0.05 was adopted (* *p* ≤ 0.05, ** *p* ≤ 0.01,). PRG4 was always set to 100 and ratios of changes of the different morphology analyses are shown (**B**–**G**).

**Figure 7 ijms-23-13007-f007:**
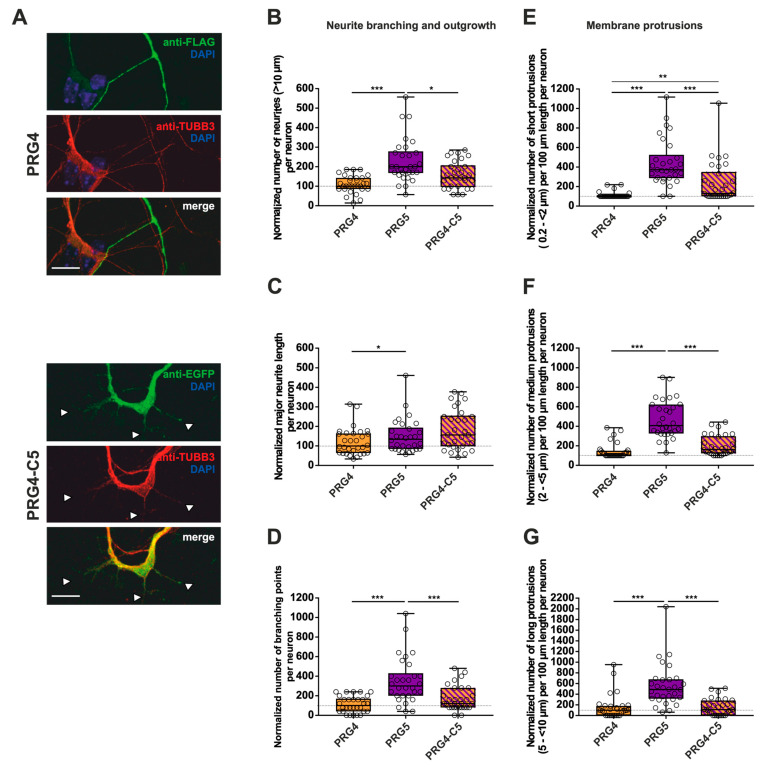
The PRG5 C-terminal domain is important in the early development of hippocampal neurons, but not sufficient to restore the full function of PRG5 when fused to PRG4. Subcellular distribution of PRG4- and PRG4-C5-encoding fusion proteins in immature primary hippocampal neurons in culture (note that the images of PRG4 are the same as in Figure 6A for better comparison). Hippocampal neurons in culture were transfected at DIV1 with (**A**) either PRG4- (upper panel) or PRG4-C5-encoding plasmids (lower panel) and analyzed at DIV2. Confocal stacks of representative neurons with immunocytochemistry of the respective fusion proteins stained for FLAG (green, PRG4) or GFP (green, PRG4-C5) to visualize the morphology of the cell or tubulin are shown. Arrowheads denote areas showing membrane localization. The scale bar represents 10 µm. Boxplots showing the quantification of (**B**) the total number of neurites (*** *p* ≤ 0.0001, * *p* ≤ 0.01); (C) the quantification of major neurite length (* *p* ≤ 0.01), and (**D**) the number of branching points (*** *p* ≤ 0.001) in PRG4- PRG5- and PRG4-C5-overexpressing neurons. Quantification of membrane protrusions in subgroups based on length in the (**E**) group of short protrusions (0.2–<2 µm) (*** *p* ≤ 0.0001; ** *p* ≤ 0.001) and of (**F**) medium protrusions (2–<5 µm) and (**G**) long protrusions (5–<10 µm) (*** *p* ≤ 0.0001). (PRG4 *n* = 30, PRG5 *n* = 30, PRG4-C5 *n* = 30 neurons). All data are presented as median and IQR. Boxes range from 1st to 3rd quartile. Horizontal lines represent medians. Whiskers represent minimum and maximum. Dots represent the values of individual neurons. Statistical analysis was performed using the Kruskal–Wallis test. A level of confidence of *p* ≤ 0.05 was adopted (* *p* ≤ 0.05, ** *p* ≤ 0.01, *** *p* ≤ 0.001). PRG4 was always set to 100 and ratios of changes of the different morphology analyses are shown (**B**–**G**).

**Table 1 ijms-23-13007-t001:** Summary of *p* values of pairwise comparisons obtained by the Kruskal–Wallis test followed by a post hoc Dunn’s test of data presented in Figure 3B–G.

Independent Variable	Group 2			
	Group 1	PRG3	PRG3∆C	PRG3-C5
Number of neurites (>10 µm) per neuron	PRG3		<0.0001	1.000
	PRG3∆C	<0.0001		<0.0001
	PRG3-C5	1.000	<0.0001	
Number of major neurite lengths per neuron	PRG3		<0.0001	1.000
	PRG3∆C	<0.0001		<0.0001
	PRG3-C5	1.000	<0.0001	
Number of branching points per neuron	PRG3		<0.0001	1.000
	PRG3∆C	<0.0001		<0.0001
	PRG3-C5	1.000	<0.0001	
Number of short protrusions (0.2–<2 µm) per 100 µm length per neuron	PRG3		<0.0001	1.000
	PRG3∆C	<0.0001		<0.0001
	PRG3-C5	1.000	<0.0001	
Number of medium protrusions (2–<5 µm) per 100 µm length per neuron	PRG3		<0.0001	0.373
	PRG3∆C	<0.0001		<0.0001
	PRG3-C5	0.373	<0.0001	
Number of long protrusions (5–<10 µm) per 100 µm length per neuron	PRG3		<0.0001	0.699
	PRG3∆C	<0.0001		0.003
	PRG3-C5	0.699	0.003	

**Table 2 ijms-23-13007-t002:** Summary of *p* values of pairwise comparisons obtained by Kruskal–Wallis test followed by a post hoc Dunn’s test of data presented in Figure 4B–G.

Independent Variable	Group 2			
	Group 1	PRG5	PRG5∆C	PRG5-C3
Number of neurites (>10 µm) per neuron	PRG5	n.s.	n.s.	n.s.
	PRG5∆C			
	PRG5-C3			
Number of major neurite lengths per neuron	PRG5		0.042	1.000
	PRG5∆C	0.042		0.014
	PRG5-C3	1.000	0.014	
Number of branching points per neuron	PRG5		0.013	1.000
	PRG5∆C	0.013		0.061
	PRG5-C3	1.000	0.061	
Number of short protrusions (0.2–<2 µm) per 100 µm length per neuron	PRG5		<0.0001	1.000
	PRG5∆C	<0.0001		<0.0001
	PRG5-C3	1.000	<0.0001	
Number of medium protrusions (2–<5 µm) per 100 µm length per neuron	PRG5		<0.0001	0.076
	PRG5∆C	<0.0001		<0.0001
	PRG5-C3	0.076	<0.0001	
Number of long protrusions (5–<10 µm) per 100 µm length per neuron	PRG5		<0.0001	1.000
	PRG5∆C	<0.0001		<0.0001
	PRG5-C3	1.000	<0.0001	

**Table 3 ijms-23-13007-t003:** Summary of *p* values of pairwise comparisons obtained by the Kruskal–Wallis test followed by a post hoc Dunn’s test of data presented in Figure 6B–G.

Independent Variable	Group 2			
	Group 1	PRG4	PRG3	PRG4-C3
Number of neurites (>10 µm) per neuron	PRG4	n.s.	n.s.	n.s.
	PRG3			
	PRG4-C3			
Number of major neurite lengths per neuron	PRG4	n.s.	n.s	n.s.
	PRG3			
	PRG4-C3			
Number of branching points per neuron	PRG4	n.s.	n.s.	n.s.
	PRG3			
	PRG4-C3			
Number of short protrusions (0.2–<2 µm) per 100 µm length per neuron	PRG4		0.023	0.502
	PRG3	0.023		0.597
	PRG4-C3	0.502	0.597	
Number of medium protrusions (2–<5 µm) per 100 µm length per neuron	PRG4		0.002	1.000
	PRG3	0.002		0.005
	PRG4-C3	1.000	0.005	
Number of long protrusions (5–<10 µm) per 100 µm length per neuron	PRG4		0.007	1.000
	PRG3	0.007		0.010
	PRG4-C3	1.000	0.010	

**Table 4 ijms-23-13007-t004:** Summary of *p* values of pairwise comparisons obtained by the Kruskal–Wallis test followed by a post hoc Dunn’s test of data presented in Figure 7B–G.

Independent Variable	Group 2			
	Group 1	PRG4	PRG5	PRG4-C5
Number of neurites (>10 µm) per neuron	PRG4		<0.0001	0.052
	PRG5	<0.0001		0.020
	PRG4-C5	0.052	0.020	
Number of major neurite lengths per neuron	PRG4		0.516	0.040
	PRG5	0.516		0.805
	PRG4-C5	0.040	0.805	
Number of branching points per neuron	PRG4		<0.0001	0.314
	PRG5	<0.0001		0.002
	PRG4-C5	0.314	0.002	
Number of short protrusions (0.2–<2 µm) per 100 µm length per neuron	PRG4		<0.0001	0.005
	PRG5	<0.0001		0.001
	PRG4-C5	0.005	0.001	
Number of medium protrusions (2–<5 µm) per 100 µm length per neuron	PRG4		<0.0001	0.098
	PRG5	<0.0001		<0.0001
	PRG4-C5	0.098	<0.0001	
Number of long protrusions (5–<10 µm) per 100 µm length per neuron	PRG4		<0.0001	1.000
	PRG5	<0.0001		<0.0001
	PRG4-C5	1.000	<0.0001	

**Table 5 ijms-23-13007-t005:** Plasmids and fusion proteins used.

Plasmid	Encoded Protein	Abbreviation	Source
pEGFP-N1	EGFP	EGFP	Clontech/Takara Holdings Shimogyo-ku, Kyoto, Japan
pEGFP-N1-mPRG3	mPRG3-EGFP	PRG3	[9,14]
pEGFP-N1-mPRG3ΔC	mPRG3ΔC-EGFP	PRG3ΔC	This paper
pEGFP-N1-mPRG3Δ-C5	mPRG3ΔC-C5-EGFP	PRG3-C5	This paper
pEGFP-N1-rPRG5	rPRG5-EGFP	PRG5	[7,8]
pEGFP-N1-rPRG5ΔC	rPRG5ΔC-EGFP	PRG5ΔC	This paper
pEGFP-N1-rPRG5Δ-C3	rPRG5Δ-C3-EGFP	PRG5-C3	This paper
pFLAG-CMV2-mPRG4	mPRG4-FLAG	PRG4	[8,14]
pEGFP-N1-mPRG4ΔC-C3	mPRG4ΔC-C3-EGFP	PRG4-C3	This paper
pEGFP-N1-mPRG4ΔC-C5	mPRG4ΔC-C5-EGFP	PRG4-C5	This paper

## Data Availability

The data analyzed during the current study are available from the corresponding author on request.

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
