# Peer review of "PRG3 and PRG5 C-Termini: Important Players in Early Neuronal Differentiation"

_ijms, 2022, doi:10.3390/ijms232113007_

Round 1
Reviewer 1 Report
This manuscript by Brandt N and colleagues would like to deeper investigate the known role of PRGs proteins in neuronal branching during development. In particular, the study has proven that the C-terminus of these membrane factors are important for their functional activity, together with other unknown domains. In my opinion, the manuscript is feasible for acceptance but it requires some clarifications:
- It does not appear clear to me why authors use non-neuronal cells to make their in vitro experiments, together with immature hippocampal neurons.
- Statistical analysis of data has shown that they are not normally distributed, so a non parametric test has been used correctly. However, if data do not follow the normal distribution, then it would be appropriate to use the median instead of the mean, as mean and standard error of the mean are parameters representing a normal distribution.
Author Response
Dear Reviewer 1,
we have carefully considered your comments. We have addressed all of the concerns you have raised. This resulted in a thoroughly revised version of our manuscript. We hope that the sum of these changes will make the paper acceptable for publication in IJMS.
Please find a point-by-point description of the changes in accordance to your comments, below:
This manuscript by Brandt N and colleagues would like to deeper investigate the known role of PRGs proteins in neuronal branching during development. In particular, the study has proven that the C-terminus of these membrane factors are important for their functional activity, together with other unknown domains. In my opinion, the manuscript is feasible for acceptance but it requires some clarifications:
- It does not appear clear to me why authors use non-neuronal cells to make their in vitro experiments, together with immature hippocampal neurons.
Response: 1 We followed the expert’s advice and addressed this issue in the „Results“ section in the revised version. Non-neuronal cells were used to test whether a neuronal protein is capable of altering the morphology of these cells, what it does in fact. The generated fusion proteins seem to generally influence filopodia formation, pointing to a more general mechanism. See L: 134-137
- Statistical analysis of data has shown that they are not normally distributed, so a non parametric test has been used correctly. However, if data do not follow the normal distribution, then it would be appropriate to use the median instead of the mean, as mean and standard error of the mean are parameters representing a normal distribution.
Response: We thank the reviewer for the comment and followed the advice. To this end we revised the data presentation thoroughly and in a way that figures and data have a good matching in the revised version. All data are shown as median/interquartile range instead of the mean and standard error. Figures 3,4, 6 and 7 and Supplementary Material Tables S1, S2, S3, S4 were changed accordingly. See also L:174, 179,192, 194, 199, 242, 245, 249, 259, 264, 524
We wish to thank the reviewer for his/her helpful comments
Yours sincerely
Prof. Dr. Anja U. Bräuer
Reviewer 2 Report
In the presented manuscript, the authors claim that plasticity-related genes (PRG3 and PRG5) promote different morphological changes in neurons, by analysis and quantification of fluorescent confocal microscopy images. The results are novel, they potentially merit publishing, once their statistical significance is properly proven. The major concern is that the authors overclaim the statistical significance of the results.
The authors artificially inflate the sample size, by using in the statitical tests n= the number of neurons, (around n=40) instead of N=the number of replicates (N=3 in their case). The statistical tests, on the ground of which, the authors claim the statistical significance, assert that the samples are mutually independent. It means, that no two observations in a dataset are related to each other, or they can affect each other in any way. This is clearly not the case here, since n=the number of neurons, is taken as the number of independent observations. The neurons were imaged on the same cover slips, they underwent the same transfection process and there were several other potential experimental factors violating the independence assumptions, see for example Lazic SE, Clarke-Williams CJ, Munafò MR. “What exactly is 'N' in cell culture and animal experiments?” PLoS Biol. 2018 Apr.
I suggest therefore performing 2-3 more independent replicates of the experiment, and recalculate the statistics with N=5 or 6, being the number of replicates.
Minor comments:
Fig. 2 and 5: The quality of the panels (the resolution) is low and it is impossible to see several protrusions indicated by the arrows. Probably, the resolution was lost somewhere during the PDF file creation, please make sure that high quality images are embedded.
It looks like that the cells with very different morphology/different part of the cells were selected (especially in Fig. 5), so it is not clear if the morphological changes observed are the result of selection of cells with different overall morphology/different parts of the cell.
Tables 2,4,6,8 are more appropriate to the Supplementary Materials, rather than main text.
Table 1 is at the very end of the manuscript, please number the tables in the order of appearance.
Why the authors display everywhere more digits for the SEM, than for the related mean value (e.g. 141.82±11.044) ?
Author Response
Dear Reviewer 2,
we have carefully considered your comments. We have addressed all of the concerns you have raised. This resulted in a thoroughly revised version of our manuscript. We hope that the sum of these changes will make the paper acceptable for publication in IJMS.
Please find a point-by-point description of the changes in accordance to your comments, below:
In the presented manuscript, the authors claim that plasticity-related genes (PRG3 and PRG5) promote different morphological changes in neurons, by analysis and quantification of fluorescent confocal microscopy images. The results are novel, they potentially merit publishing, once their statistical significance is properly proven. The major concern is that the authors overclaim the statistical significance of the results.
The authors artificially inflate the sample size, by using in the statitical tests n= the number of neurons, (around n=40) instead of N=the number of replicates (N=3 in their case). The statistical tests, on the ground of which, the authors claim the statistical significance, assert that the samples are mutually independent. It means, that no two observations in a dataset are related to each other, or they can affect each other in any way. This is clearly not the case here, since n=the number of neurons, is taken as the number of independent observations. The neurons were imaged on the same cover slips, they underwent the same transfection process and there were several other potential experimental factors violating the independence assumptions, see for example Lazic SE, Clarke-Williams CJ, Munafò MR. “What exactly is 'N' in cell culture and animal experiments?” PLoS Biol. 2018 Apr.
I suggest therefore performing 2-3 more independent replicates of the experiment, and recalculate the statistics with N=5 or 6, being the number of replicates.
Response: We thank the reviewer for the suggestions and clarified this issue in the „Materials and Methods“ Section and included more details for better understanding. „More specifically, for each independent experiment, hippocampi from all embryos (7-9 on average) of a pregnant mouse were collected and pooled. For each fusion protein, at least two to three coverslips were imaged per independent experiment to achieve higher variability, resulting in a total of n=10-15 neurons“. Therefore, the neurons underwent different transfection processes per condition and neurons were imaged on different coverslips to avoid a bias. See L: 508-511
Minor comments:
Fig. 2 and 5: The quality of the panels (the resolution) is low and it is impossible to see several protrusions indicated by the arrows. Probably, the resolution was lost somewhere during the PDF file creation, please make sure that high quality images are embedded.
Response: We followed the reviewer’s advice and adjusted the quality of our images carefully.
It looks like that the cells with very different morphology/different part of the cells were selected (especially in Fig. 5), so it is not clear if the morphological changes observed are the result of selection of cells with different overall morphology/different parts of the cell.
Response: We thank the reviewer for the comment. We know that HEK cells induce filopodia outgrowth (morphological changes) due to overexpression of PRG3 and PRG5, and have been able to show that this does not occur after overexpression of PRG4. We have recently quantified and published this (Gross et al., 2021). For the figures, we have chosen representative cells reflecting the morphological changes due to overexpression of the respective fusion proteins. We hope that clarifies the concern raised by the reviewer.
Tables 2,4,6,8 are more appropriate to the Supplementary Materials, rather than main text.
Response: As suggested, we have moved the Tables 2, 4, 6, 8 to the Supplementary materials (now referred to as Table S1, S2, S3, S4).
Table 1 is at the very end of the manuscript, please number the tables in the order of appearance.
Response: We agree with the reviewer and changed this accordingly.
Why the authors display everywhere more digits for the SEM, than for the related mean value (e.g. 141.82±11.044)?
Response: We thank the Reviewer for the advice and addressed this issue in the revised version.
We wish to thank the reviewer for his/her helpful comments
Yours sincerely
Prof. Dr. Anja U. Bräuer
Reviewer 3 Report
This is an elegant study about the implications of the PRG3 and PRG5 proteins' C-terminal domain in promoting morphological extension in neuroblasts during brain development. By modifying these regions, the authors reveal the role of those protein fragments in the localization in the plasma membrane and their role in neurites extension.
However, I miss a functional or adaptive explanation for conducting these experiments, both in the introduction and in the discussion. That physiological implication would give these results higher significance and soundness to society.
The figures are of high quality. By contrast, I consider that the figure feet are too extensive and give too much information, some of which would be better included in the main text of the results.
The technical information is adequate. However, the quantity of protein loaded in every western blot is missing and must be added.

Author Response
Dear Reviewer 3,
we have carefully considered your comments. We have addressed all of the concerns you have raised. This resulted in a thoroughly revised version of our manuscript. We hope that the sum of these changes will make the paper acceptable for publication in IJMS.
Please find a point-by-point description of the changes in accordance to your comments, below:
This is an elegant study about the implications of the PRG3 and PRG5 proteins' C-terminal domain in promoting morphological extension in neuroblasts during brain development. By modifying these regions, the authors reveal the role of those protein fragments in the localization in the plasma membrane and their role in neurites extension.
However, I miss a functional or adaptive explanation for conducting these experiments, both in the introduction and in the discussion. That physiological implication would give these results higher significance and soundness to society.
Response: According to the suggestion of the reviewer more details of our aim concerning a functional explanation were included in the „Introduction“ and „Discussion“ sections. We agree with the reviewer that this helps to understand the significance of our study. See L: 98-104; 358-366.
The figures are of high quality. By contrast, I consider that the figure feet are too extensive and give too much information, some of which would be better included in the main text of the results.
Response: We followed the expert’s advice that the figure feets were too extensive and addressed the issue in the revised version. We shortened the figure feets.
The technical information is adequate. However, the quantity of protein loaded in every western blot is missing and must be added.
Response: We thank the reviewer for the comment. However, it was not the goal to quantify the protein amount, as we only wanted to prove that our constructed fusion proteins were expressed in the cells. Our intention was to detect the expression of our created fusion proteins by immunoprecipitation in the cells. Therefore, we used purified recombinant fusion proteins for our experiments. In our western blot, we show that our newly generated fusion proteins are expressed in cells and are not degraded. We hope this clarifies the issue.
We wish to thank the reviewer for his/her helpful comments
Yours sincerely
Prof. Dr. Anja U. Bräuer